# Characteristics of Soy Protein Prepared Using an Aqueous Ethanol Washing Process

**DOI:** 10.3390/foods10092222

**Published:** 2021-09-18

**Authors:** Yu Peng, Konstantina Kyriakopoulou, Mbalo Ndiaye, Marine Bianeis, Julia K. Keppler, Atze Jan van der Goot

**Affiliations:** 1Food Process Engineering Group, Wageningen University & Research, P.O. Box 17, 6700 AA Wageningen, The Netherlands; yu1.peng@wur.nl (Y.P.); julia.keppler@wur.nl (J.K.K.); atzejan.vandergoot@wur.nl (A.J.v.d.G.); 2R&D Department, Avril Groupe, 35170 Bruz, France; mbalo.ndiaye@groupeavril.com (M.N.); marine.bianeis@groupeavril.com (M.B.)

**Keywords:** hydro-ethanol washing, soybean protein, cold pressing, oil-containing ingredients, free sugars, soy protein concentrate, fractionation

## Abstract

Currently, the predominant process for soy protein concentrate (SPC) production is aqueous ethanol washing of hexane-extracted soy meal. However, the use of hexane is less desired, which explains the increased interest in cold pressing for oil removal. In this study, cold-pressed soy meal was used as the starting material, and a range of water/ethanol ratios was applied for the washing process to produce SPCs. Washing enriched the protein content for the SPCs, regardless of the solvent used. However, we conclude that washing with water (0% ethanol) or solvents with a high water/ethanol ratio (60% and above) can be more advantageous. Washing with a high water/ethanol ratio resulted in the highest yield, and SPCs with the highest protein solubility and water holding capacity. The water-only washed SPC showed the highest viscosity, and formed gels with the highest gel strength and hardness among all the SPCs at a similar protein concentration. The variations in the functionality among the SPCs were attributed to protein changes, although the effects of non-protein constituents such as sugar and oil might also be important. Overall, the aqueous ethanol washing process combined with cold-pressed soy meal created SPCs comparable to commercial SPC in terms of composition, but with varied functionalities that are relevant for novel soy-food developments.

## 1. Introduction

Soybean has been recognized as a powerhouse of nutrients, rich in protein and oil, as well as carbohydrates [1]. After fractionation, various protein-rich ingredients can be obtained from it, which have gained great popularity due to their high nutritional value and versatile functional properties [2]. Three forms of soy protein-rich ingredients are commercially available according to the protein content (N × 6.25), namely soy flour (50–65%), soy protein concentrate (SPC, 65–90%), and soy protein isolate (SPI, >90%) [3]. Among them, SPC was the last one to be developed; however, it nowadays provides a broader application range than soy flour, and is less expensive than SPI [4]. While the production of SPI uses alkaline conditions to maximally solubilize protein, SPC is made under conditions where the majority of the soy proteins become insoluble allowing the non-protein constituents to be washed away [5]. Different washing processes have been developed for the production of SPCs, which have included: water washing of heat-denatured soy meal, acid washing at the isoelectric pH, and aqueous ethanol washing [6]. Aqueous ethanol washing gradually has become the predominant process because high yields can be attained, and the obtained SPC is relatively bland in taste with less beany flavors [7]. Moreover, the process has been found to eliminate the allergenicity of soy proteins [8].

Aqueous ethanol washing is based on the ability of aqueous ethanol solutions to extract soluble fractions such as free sugars and other low molecular weight constituents of defatted soy meal (DFSM) without solubilizing the soy protein. DFSM, a by-product of the soy-oil production, is usually created after hexane extraction of soybeans. The use of hexane leads to extraction yields of crude oil greater than 95% [9]. However, hexane is highly explosive and, nowadays, is considered to be toxic. The concerns about its safety and emission restrictions have stimulated interest in alternative defatting techniques [10]. Among those, cold pressing is regarded as a rapid and environmentally friendly technique without using the toxic solvents [11]. In addition, the relatively low operation temperature can preserve the protein nativity [12]. Nevertheless, cold pressing is not as efficient for oil extraction as the hexane extraction.

The studies to date have focused on using an aqueous ethanol washing process to produce SPCs with low contents of residual oil and sugars, while hexane-extracted soy meal is commonly used as the starting material. To the best of our knowledge, limited information is available on the use of aqueous ethanol washing in combination with cold-pressed soy meal, especially the effect of water/ethanol ratios on the overall composition, and thus the functionality of SPCs. Therefore, in this study we used cold-pressed DFSM as the starting material instead of hexane-extracted DFSM. In addition, the potential to use a water-only washing process is not sufficiently described as a way to create SPCs under a clean label. We hypothesize that the protein, oil, and carbohydrate content of the produced SPCs, as well as the final functionality can be tuned by choosing specific water/ethanol ratios. To prove this, we used aqueous ethanol washing with different water/ethanol ratios (0–100%) to produce SPCs with varying composition. The composition, yield, and functional properties such as solubility, water holding capacity, and thermal stability of the SPCs were investigated. The rheological properties of the SPCs were also evaluated with multiple protein concentrations (6, 9, and 15 wt%). Since the developed process leaves some oil in the SPCs, we also measured the primary and secondary oil oxidation products to evaluate the oil-containing soy protein ingredients. The objective of this study was to examine the current process for commercial SPC and adjust it for producing hexane-free or even completely organic solvent-free SPCs with a range of compositions and functionalities.

## 2. Materials and Methods

### 2.1. Materials

Defatted soy meal (DFSM) obtained after mechanical cold pressing was provided by the Avril group (France). No organic solvent was used for the defatting treatment according to the supplier. The DFSM contained 43.1 ± 0.6% of protein and 10.3 ± 0.9% of oil on a dry basis. The commercial SPC was obtained from Solae (Europe S.A.).

For the aqueous ethanol washing and composition analysis, 96% ethanol and petroleum ether were purchased from EMD Millipore Corporation (Merck KGaA, Darmstadt, Germany). For the phenol content analysis, Folin–Ciocalteau (FC) reagent was obtained from MP Biomedicals (MP Biomedicals, Illkirch Cedex, France). Sodium carbonate (anhydrous ≥99.5% ACS) was obtained from VWR International (VWR International GmbH, Darmstadt, Germany).

For the oil oxidation analysis, iron(II) sulphate heptahydrate (FeSO_4_⋅7H_2_O), cumene hydroperoxide solution (80%), n-hexane, sodium chloride (NaCl), and para-anisidine were purchased from Sigma-Aldrich (Sigma-Aldrich, Burlington, VT, USA); hydrochloric acid (37%), acetic acid (glacial), barium chloride dihydrate (BaCl_2_⋅2H_2_O), ammonium thiocyanate (NH_4_SCN), 2-propanol, and 1-butanol were obtained from Merck Millipore (Merck KGaA, Darmstadt, Germany); methanol was purchased from Actu-All Chemicals B.V. (Actuall Chemical, Oss, The Netherlands).

The ultrapure water used in this study was purified with a Milli-Q Lab Water System (Milli-Q IQ 7000 Ultrapure Lab Water System, Merck KGaA, Darmstadt, Germany).

### 2.2. Aqueous Ethanol Washing Process

A scheme for the aqueous ethanol washing process is provided in Figure 1. Aqueous ethanol solutions were prepared in advance by mixing water with 96% ethanol in a Schott glass bottle with the correct volume. The ethanol to the total solvent volume ratio is referred to, in this study, as water/ethanol ratios of 0%, 20%, 40%, 60%, 80%, and 100%.

DFSM was mixed with aqueous ethanol solution in a solid/liquid ratio of 1:10 (*w*/*v*) and stirred at room temperature (25 °C) for 30 min. Then, the dispersion was centrifuged (20,000× *g*, 30 min, 25 °C) to separate the supernatant and pellet. The supernatant was collected as the extract, and the pellet was transferred to a fume hood overnight to evaporate the ethanol. Subsequently, the pellet was freeze-dried (Freeze Dryer, Martin Christ, Osterode, Germany) and milled into powder, which is reported in this study as dried soy protein concentrates (SPCs). All the extracts and SPCs processed with varied water/ethanol ratios were prepared in triplicate and stored at 4 °C for further analysis.

### 2.3. Microscopic Analysis

Scanning electron microscope (SEM, Phenom Pure G2, Phenom-world BV, Eindhoven, The Netherlands) was performed for viewing the microstructures of the SPC powders. The SPC was evenly placed on an aluminum sample holder using double-sided adhesive conductive carbon tape, and the microstructure was observed with an accelerating voltage at 5 kV.

### 2.4. Composition and Yields

The protein contents of the DFSM and SPCs were determined by using a nitrogen analyzer (Flash EA 1112 series Dumas, Thermo Scientific, Breda, The Netherlands). A nitrogen-to-protein conversion factor of 5.7 was used [13]. The oil content was determined by using petroleum ether as an extraction solvent with a Buchi extraction system B-811LSV (Buchi Labortechnik AG, Flawil, Switzerland). The sugar content was determined with a Raffinose Sucrose D-Glucose (K-RAFGL) assay from Megazyme (Megazyme International Ireland, Wicklow, Ireland) and the method was adapted from McCleary et al. [14]. The total free sugars were the sum total amount of glucose, sucrose, and galactosyl-sucrose oligosaccharides (GOS). All measurements were performed in triplicate.

The dry matter yield was calculated using Equation (1), while the protein yield was calculated according to Equation (2):(1)$$\mathrm{Dry}\;\mathrm{matter}\;\mathrm{yield}\;\left(\%\right)=\frac{\mathrm{Weight}\;\mathrm{of}\;\mathrm{dried}\;\mathrm{SPC}}{\mathrm{Weight}\;\mathrm{of}\;\mathrm{starting}\;\mathrm{DFSM}}\times 100\%\;$$
(2)$$\mathrm{Protein}\;\mathrm{yield}\;\left(\%\right)=\frac{\mathrm{Total}\;\mathrm{protein}\;\mathrm{in}\;\mathrm{dried}\;\mathrm{SPC}}{\mathrm{Total}\;\mathrm{protein}\;\mathrm{in}\;\mathrm{starting}\;\mathrm{DFSM}\;}\times 100\%\;$$

### 2.5. Protein Molecule Profile with SDS-PAGE

The protein molecular profile of all the SPCs and commercial SPC was characterized by SDS-PAGE. The SPC samples were prepared with water to reach a concentration of 2 mg protein per mL in a falcon tube and were centrifuged (20,000× *g*, 30 min, 25 °C). Afterwards, the insoluble parts were removed, and the supernatant of each fraction was used for the SDS-PAGE under reducing condition. Approximately 20 µL of each protein supernatant was mixed well with 1 µL β-mercaptoethanol and 20 µL Bio-Rad sample buffer (62.5 mM Tris-HCl, 25% *v*/*v* glycerol, 4% *w*/*v* SDS, and 0.01% *w*/*v* bromophenol blue) and heated at 95 °C for 10 min. After heating, the sample was centrifuged at 10,000× *g* for 1 min. Precast gel (Mini-PROTEAN TGX, Bio-Rad, Lunteren, The Netherlands) was used, while 5 μL of SDS-PAGE marker and 15 μL of the sample were loaded onto the gel wheels. The electrophoresis was performed at 200 V in a Mini-Protean II electrophoresis cell (Bio-Rad, Veenendaal, The Netherlands) with the diluted Bio-Rad running buffer (0.025 M Tris, 0.19 M glycine, and 0.1% SDS). After approximately 40 min, the gels obtained after electrophoresis were stained using Bio-Safe Coomassie Blue for approximately 1 h, and washed with water multiple times before transferring to gel scanner (Biorad-GS900, Veenendaal, The Netherlands).

### 2.6. Oil Oxidation: Primary and Secondary Products

The oil oxidation levels in the DFSM and SPCs were analyzed by measuring the primary (hydroperoxides concentration) and secondary (aldehydes) reaction products.

#### 2.6.1. Hydroperoxide Concentration

The hydroperoxide concentration (i.e., primary oxidation product) was measured according to the method described by Shanta et al. [15] and Feng et al. [16] with slight modifications. Approximately 0.3 g of DFSM or SPCs was mixed with 1.5 mL n-hexane/2-propanol (3:1, *v*/*v*), vortexed 3 × 10 s with 20 s intervals, and centrifuged at 10,000× *g* for 2 min. Subsequently, 0.2 mL of the supernatant was mixed well with 2.8 mL of the methanol/1-butanol (2:1, *v*/*v*) and 3.94 M ammonium thiocyanate/ferrous iron solution (1:1, *v*/*v*; 30 μL) for the reaction. After 20 min, the absorbance was measured at 510 nm with a DU 720 UV-visible spectrophotometer (Beckman Coulter, Woerden, The Netherlands). The hydroperoxide concentration was calculated using a cumene hydroperoxide standard curve.

#### 2.6.2. Para-Anisidine Value (pAV)

Total aldehyde content (i.e., a secondary oxidation product) was indicated by the para-anisidine value (pAV), using the AOCS official method [17]. Approximately 1 g of DFSM or SPCs was mixed with 1 mL saturated sodium chloride solution and 5 mL n-hexane/2-propanol (3:1, *v*/*v*), vortexed 3 × 10 s with 20 s intervals and centrifuged at 2000× *g* for 8 min. The absorbance of 1 mL of the supernatant (Ab) was measured at 350 nm using hexane as a blank. Subsequently, 1 mL of supernatant or hexane was mixed with 0.2 mL of 2.5 g/L para-anisidine in acetic acid solution. After 10 min, the absorbance (As) was measured at 350 nm, using hexane with para-anisidine solution as a blank. The pAV was calculated using Equation (3), and the m is the mass of oil per mL hexane (g/mL):pAV = (1.2 × As − Ab)/m(3)

### 2.7. Total Phenolic Content (TPC) in the Extract

The total phenol content (TPC) in the extract was determined by using the Folin–Ciocalteau (FC) method described by Singleton, Orthofer, and Lamuela-Raventós [18] with slight modifications. Approximately 100 µL of sample taken from each extract was added to 7.9 mL of water and mixed well. Subsequently, 500 µL of FC reagent and 1.5 mL of 20% (*w*/*v*) sodium carbonate were added, mixed thoroughly using a vortex mixer, and heated at 40 °C for 30 min. The absorbance was measured at 750 nm with a spectrophotometer (DR3900 Laboratory VIS Spectrophotometer, Hach, Loveland, CO, USA). Different aqueous ethanol ratios were used as the blank. The TPC was calculated based on a calibration curve plotted with gallic acid (0.025 to 4 mg/mL) as a standard. The results were expressed as mg of gallic acid equivalents (GAE) per mL of extract (mg GAE/mL extract).

### 2.8. Particle Size Distribution

The particle size distribution (PSD) of the SPCs was measured with a laser light scattering instrument (Mastersizer 3000, Malvern Instruments Ltd., Malvern, UK) and a wet module (Hydro SM, Malvern Instruments Ltd., Malvern, UK). For these measurements, 1% (*w*/*v*) protein dispersions were prepared with Milli-Q water. A refractive index of 1.45 was used for the dispersion phase and 1.33 for the water continuous phase.

### 2.9. Differential Scanning Calorimetry (DSC)

The protein denaturation temperature (T*_d_*) and the enthalpy of the transition of the SPCs were measured by differential scanning calorimetry (Diamond DSC, PerkinElmer, Waltham, MA, USA); 20% (*w*/*w*) protein dispersions of each SPC were prepared in high-volume aluminum pans, and then well sealed. The DSC was calibrated with indium, and an empty aluminum pan was used as the reference. The sample was scanned at 5 °C/min from 20 °C to 150 °C. Measurements were analyzed with Start Pyris Software.

### 2.10. Nitrogen Solubility Index (NSI)

The nitrogen solubility index (NSI) is routinely used to evaluate protein solubility [19]. A 2% (*w*/*w*) dispersion of each SPC sample was placed in a centrifuge tube and moderately shaken overnight. Subsequently, the sample was centrifuged at 18,670× *g* for 30 min to separate the supernatant and pellet. The pellet was oven-dried at 105 °C and weighed. The nitrogen contents in the oven-dried pellets were measured by using the Dumas analysis. The NSI was calculated by the ratio of soluble nitrogen over the total initial nitrogen content present in the SPC. All the measurements were performed in triplicate.

### 2.11. Water Holding Capacity (WHC) and Oil Absorption Capacity (OAC)

The water holding capacity (WHC) of the SPC was measured using the method described by Peters [20] with slight modifications. A 2% (*w*/*v*) dispersion was prepared with water and was stirred overnight. Then, the dispersion was centrifuged at 20,000× *g* for 30 min to separate the supernatant and pellet. The wet pellet was collected, weighed and oven-dried at 105 °C overnight. The dry pellet was obtained and weighed again. The WHC was calculated using Equation (4), while the water holding capacity of the pellet (WHC_P_) was calculated using Equation (5):(4)$$\mathrm{WHC}\;\left(\frac{g\;\mathrm{water}}{g\;\mathrm{dry}\;\mathrm{SPC}\;}\right)=\frac{M_{\mathrm{wet}\;\mathrm{pellet}}-M_{\mathrm{dry}\;\mathrm{pellet}}}{M_{\mathrm{dry}\;\mathrm{SPC}}}$$
(5)$${\mathrm{WHC}}_{P}\;\left(\frac{g\;\mathrm{water}}{g\;\mathrm{dry}\;\mathrm{pellet}}\right)=\frac{M_{\mathrm{wet}\;\mathrm{pellet}}-M_{\mathrm{dry}\;\mathrm{pellet}}}{M_{\mathrm{dry}\;\mathrm{pellet}}}$$

The oil absorption capacity (OAC) was measured using the method described by Lin [21] with slight modifications. Approximately 0.5 g of the SPC (dry basis) and 10.0 mL of rapeseed oil were added to a 15 mL conical graduated centrifuge tube and mixed for 3 min with a vortex mixer to disperse the sample into the oil. After a holding period of 30 min, the tube was centrifuged for 25 min at 3050× *g*. Then, the separated oil was removed with a pipette, and the tube was inverted to drain the unbound oil prior to reweighing. The OAC was expressed using Equation (6):(6)$$\mathrm{OAC}\;\left(\frac{g\;\mathrm{oil}}{g\;\mathrm{dry}\;\mathrm{SPC}\;}\right)=\frac{M_{\mathrm{pellet}}-M_{\mathrm{dry}\;\mathrm{SPC}}}{M_{\mathrm{dry}\;\mathrm{SPC}}}$$

### 2.12. Viscosity Analysis

Soy protein dispersions were prepared and stirred for approximately 1 h before the viscosity measurement. The protein concentrations for all the dispersions were standardized at 6 wt%, according to different protein contents in the SPCs (Table 1). A stress-controlled rheometer (Anton Paar Physica MCR 301, Graz, Austria) combined with a sand-blasted CC-17 concentric cylinder geometry was used to determine the viscosity. Each SPC dispersion was equilibrated for 5 min, and a shear rate sweep was performed at 25 °C from 0.1 to 100 s^−1^, and back to 0.1 s^−1^ again. One hundred data points were collected using a constant interval of 20 s between points during the measurements.

### 2.13. Gelation Behaviors

A stress-controlled rheometer (Anton Paar Physica MCR 301, Graz, Austria) combined with a sand-blasted CC-17 concentric cylinder geometry was used. The protein concentrations for all the dispersions were standardized at 9 wt%, according to different protein contents in the SPCs (Table 1). A thin layer of high-temperature-resistant silicone oil covered the top of the samples and prevented water evaporation during heating. The temperature, frequency, and strain sweeps were performed sequentially. The temperature sweep was done by increasing the temperature from 20 to 95 °C at a rate of 3 °C/min, followed by 10 min at 95 °C, before cooling to 20 °C at a rate of 3 °C/min. Subsequently, the samples were exposed to a frequency sweep from 0.01 to 10 Hz (at a strain of 1%), and a strain sweep from 0.1 to 1000% (at a frequency of 1 Hz). The storage module (G′) and loss module (G″) dependency on temperature, frequency, and strain were recorded.

### 2.14. Textural Analysis of Soy Protein Gels

To make the soy protein gels, 15 wt% protein concentration was selected based on preliminary experiments to ensure all the SPCs formed a firm gel (Table 1). SPCs were mixed with water and hydrated for 30 min before being transferred to a stainless steel gelatin vessel (internal height 5 mm and radius 12.5 mm). The vessel was hermetically sealed and heated in a water bath at 95 °C for 30 min. After heating, the vessel was cooled by using running water for 15 min and, subsequently, the gel was removed.

The texture profile analysis (TPA) was carried out on the gels using a texture analyzer (Instron, Norwood, MA, USA) under room temperature conditions. A double compression mode was exerted by a cylindrical probe with a flat section (diameter 75 mm) at a displacement speed of 1 mm/s. The compression distance was 3 mm (60%), while the relaxing time between two compressions was 30 s. The tests generated a plot of force vs. time, from which the hardness, springiness, cohesiveness, and chewiness were calculated.

### 2.15. Statistical Analysis

All the extracts and SPCs processed with varied water/ethanol ratios were prepared in triplicate and stored at 4 °C for further analysis. Data were collected for each sample from at least three experiments. IBM SPSS Statistics Version 23.0 was used to analyze the variance, and the Duncan’s test was performed to determine the statistical significance between samples at an α level of 0.05.

## 3. Results

### 3.1. Protein Content, Yield, Composition, and Microstructure

The powder morphologies of the DFSM and all the SPCs were visualized with SEM (Figure 2). Before the washing process, cold-pressed DFSM presented a globular and compact structure, with some small fragments on the surface. After washing with water (0% ethanol), the surface of the SPC was irregularly wrinkled with a spongy porous structure. This feature was possibly attributed to the predominant insoluble carbohydrates such as cellulose and hemicellulose that remained after washing [22]. Thus, this can be taken as a sign for both protein loss and removal of other soluble components. When the water/ethanol ratio was 20% and 40%, some attached spherical shapes were observed on the cratered surface of SPCs. These may be the protein bodies released from cellular matrixes [23]. The use of high water/ethanol ratios (80% and 100%) resulted in dense structures with no obvious open pores on the surface.

The obtained SPCs presented higher protein content (between 46.48 and 59.49%) than the start material DFSM (43.14%), confirming that the aqueous ethanol washing processes can enrich the protein in the ingredients (Figure 3A). With an increase in the water/ethanol ratio from 0 to 100%, the protein content of SPC increased until it reached a maximum value with 59% protein at 40% ethanol, and then decreased again. This high protein concentration approaches the measured protein content of commercial SPC (61.68%). By contrast, a protein enrichment of only 3.4% (from 43.1 to 46.5%) was achieved by washing with only water (0% ethanol). Previous studies that have applied the aqueous ethanol washing process to defatted sunflower kernels have reported a similar change in protein content with an increase in the water/ethanol ratio (0–100%) [23]. The uptrend of protein content may result from the removal of oil and sugars, while the downtrend might be due to the low polarity of ethanol, which influences the protein extractability in the solvent [24]. Washing with water also induced the SPC with the lowest dry matter yield and protein yield, which may indicate the highest loss of both protein and non-protein constituents. The highest yield was found when 80% ethanol was applied, which was approximately 81% for the total mass and 96% for the protein. By interpreting these results after taking into consideration the SEM-pictures, it seems that an open structure could be the result of protein loss. This means that the remaining insoluble carbohydrates become more visible (Figure 2).

The protein compositions of all the obtained SPCs and commercial SPC were analyzed by SDS-PAGE under reducing conditions (Figure 3B). Similar bands were observed in all the samples, which were identified as 7S α, α’, and β subunits; 11S A3 subunit; acidic and basic proteins according to the mass (kDa) [25]. The two main types of storage protein in soybean are 7S and 11S [26], and similar protein profiles have also been found in other soy products, such as soy flour [27], soymilk [28,29], SPC, and SPI [30]. In addition, washing with high levels of ethanol achieved SPCs that had more low-molecular-weight subunits and fewer high-molecular-weight proteins. This means that the average molecular weight of the protein present in the SPC decreases when washed with higher water/ethanol ratios.

### 3.2. Oil Content and Oil Oxidation

The DFSM used in this study was obtained after cold pressing, without using hexane solvents. It contained approximately 10.3% residual oil before washing, which was significantly higher than the DFSM obtained after intensive hexane extraction (~0.15%) [31]. Aqueous ethanol washing extracted part of the oil from the DFSM, and the oil content of the SPC became lower with an increase in the water/ethanol ratio (Figure 4A). The lowest oil content in an SPC was approximately 2.13% with a water/ethanol ratio of 100%, confirming that ethanol can work as an efficient oil-extracting solvent [32]. However, less than 1% of the oil was removed by water (0% ethanol) and approximately 9.41% of the oil was left in the SPC. The use of other ethanol/water ratios removed roughly half of the oil.

The oxidation level of the oil that remained in the SPC should remain low, because oxidation that is too high can produce rancidity and potential toxicity, leading to off-flavors [33], and thus limiting applicability. Moreover, oxidized oil may result in co-oxidation of protein, and thus negatively affect the protein functionality [34]. Figure 4C,D presents the primary and secondary oil oxidation products, respectively. The hydroperoxide concentration in the starting material DFSM was approximately 0.39 meq/kg of oil, which was lower than the concentration of hydroperoxide in the hexane-extracted soy oil (0.60 meq/kg) [35]. The pAV value of oil in the DFSM, which reflected the content of secondary oil oxidation products such as aldehyde content, was higher than the value of oil in the fresh full-fat soy flour (1.13 meq/kg) [36], as well as solvent-extracted soy oil (1.88 meq/kg) [35]. In general, by combining the results of PV and pAV, it indicated that the aqueous ethanol washing process increased the oil oxidation in the SPCs. Among all the water/ethanol ratios, the SPC prepared with 20% ethanol showed the highest amount of primary and secondary oil oxidation products. Oxidized oil off-odor was also observed for this sample. The oil oxidation level in all the other SPCs was relatively low and in acceptable levels according to the Codex Standard. The Codex Standard indicates that the PV for cold-pressed and virgin oil should be below 15 meq/kg [3].

There are several reasons for the high oxidation values found after washing with 20% ethanol. One cause could be the low remaining phenol content in that SPC, which reduces the antioxidant capacity of the total sample [37]. In addition, many phenolic compounds and plant phenolic extracts have been demonstrated to retard oil oxidation in different foods [38]; however, this is not a complete explanation because the use of only water as the solvent removed even more phenolic compounds (Figure 4B), while the oxidation level was much lower in this SPC. There are, however, other possible explanations. It has been reported that the activity of lipoxygenase (LOX) is highly correlated to the oil oxidation level, resulting in the generation of odor compounds in soybean and soy foods [39]. An early study investigated the effects of water/ethanol soaking on the LOX activity in soybean, and reported that water/ethanol ratios of 30–50% were more effective for inhibiting lipoxygenase than 10–30% [40]. Herein, the SPC washed with 20% ethanol may have higher LOX activity than other SPCs, leading to the highest oil oxidation level. Another explanation might be that 20% ethanol washed mostly non-oxidized oil (intact oil bodies), and therefore only the oxidized oil remained in the SPC. Further analyses should be conducted to better understand the underlying reason.

### 3.3. Total Free Sugars

Before the washing process, DFSM contained approximately 8% total free sugars (Figure 5). With an increase in the water/ethanol ratio from 0 to 100%, the free sugar content of SPC, first, became lower, and then increased as the water/ethanol ratio increased above 60%. Although water as a solvent for the removal of free sugars seems promising, the lowest value was found when 40% ethanol was used for washing. This trend was in line with previous studies in which it was reported that 50% ethanol can achieve better removal of free sugars from toasted soybean meal than 80% ethanol [41]. The sugar content remained above 5% in the SPC once the water/ethanol ratio in the solvent was 80% and higher. The SPC processed with a 100% water/ethanol ratio showed the highest free sugar content of 10.57%, even higher than the value of DFSM, which could be caused by the simultaneous removal of oil and part of the protein at this high ethanol content. During the production of commercial SPC, the free sugars are mostly removed and end up in the soy molasses as a by-product [42].

In general, the amounts of sucrose in the DFSM and SPCs were greater than the total amount of GOS, while the content of glucose was low. The impacts of the water/ethanol ratios on the sucrose and GOS contents were similar and in line with the changes of total free sugars. This means that the lowest sucrose and GOS contents were also found when 40% ethanol was applied for washing, while above 40%, the sucrose and GOS mostly remained in the SPC. Washing with a higher water/ethanol ratio generally led to an increased dry matter yield (Figure 3A), which may suggest that more free sugars were kept in the SPC. A similar trend was also reported in a previous study that described complete extraction of GOS from green peas when washed with 50% ethanol. A further increase in ethanol concentration to 80% reduced the extraction efficiency of GOS at room temperature [43].

### 3.4. Protein Denaturation in SPCs

The thermal transition characteristics (denaturation temperature T_d_ and protein denaturation enthalpy) of all the SPCs are presented in Table 2. Two denaturation peaks were detected in all the samples, which can be related to the denaturation changes of 7S and 11S proteins [44]. Previous studies reported that ethanol treatment leads to the denaturation of soy protein [45]. In addition, in the present study, the lowest degree of protein denaturation was observed when low water/ethanol ratios (0 and 20%) were applied for washing; however, at high ethanol ratios the denaturation also was not extreme. Specifically, the T_d_ of 7S in the various SPCs was between 74.6 and 76.1 °C, but the differences were not significant. The enthalpy of denaturation was altered by the use of different washing solvents. SPCs washed with water and 20% ethanol exhibited the highest enthalpy values, meaning least denaturation occurred during the washing process. The enthalpy values revealed that, on the one hand, washing with 40% and 60% ethanol induced SPCs with relatively higher degrees of denaturation in 7S. On the other hand, no significant differences in the degree of denaturation were observed in the case of 11S. Early studies found that the 7S component can be rapidly denatured when in contact with ethanol-water mixtures of 20% or greater ethanol concentration, while the rate of denaturation of the 11S was slow regardless of the ethanol concentration [46]. One possible explanation for this could be that the 7S protein underwent more conformational transition than the 11S protein after ethanol treatment [47]. Nevertheless, the variations in the T_d_ of the 11S protein in SPCs were larger than the 7S protein. The T_d_ of 11S protein ranged from 92.4 to 96.7 °C, while the SPCs prepared with 40% and 60% ethanol presented the lowest T_d_ values. Considering that for the SPCs washed with high water/ethanol ratios (80 and 100%), the sucrose mostly remained after washing (Figure 5). This may prevent the soy protein from the ethanol-induced denaturation. The presence of sucrose was found to stabilize the structure of proteins, for example, in the case of whey protein and bovine serum albumin [48,49]. This is also the reason why sugars are often added to protein solutions as lyoprotectants. As a result, among all the SPCs, the water-ethanol-washed SPCs with 40% and 60% water/ethanol ratios presented the highest level of protein denaturation. No denaturation peak was detected from the commercial SPC. This complete denaturation might be caused by the intensive aqueous ethanol washing steps, or heat treatments during the process, including spray drying for the final dry powder production.

### 3.5. Particle Size Distribution (PSD)

Figure 6 shows that the washing process using solvents with higher water/ethanol ratios led to a wider PSD range of SPC dispersions, while higher water/ethanol ratios resulted in a gradual shift in the major PSD peak towards larger sizes. This indicated the occurrence of ethanol-induced aggregation in these samples. When the water/ethanol ratio was 0%, the major PSD peak appeared with a particle size of approximately 90 µm; the size of the major peak was similar and approximately 160 µm when the water/ethanol ratios were 20% and 40%. The major peak size became approximately 600 µm when the water/ethanol ratios were 60% and above. These results are in line with those of previous studies. It has been reported that ethanol treatment of soy β-conglycinin with increasing water/ethanol ratios from 0–80% (*v*/*v*) led to a gradual shift in major PSD peaks towards greater sizes, while the greatest change occurred when the water/ethanol ratio was above 40% [50]. In addition, chicken egg albumin has been shown to undergo certain structural changes in water-ethanol solution, and the degree of aggregation was linearly depending on the ethanol concentration [51]. Therefore, we propose that ethanol-induced aggregation may be taking place between soy proteins and/or other constituents. For example, ethanol has been shown to induce changes in the conformations of globular proteins as a result of disruption of noncovalent interactions [52,53]. The addition of ethanol is generally considered to weaken noncovalent bonds in protein, including hydrogen and ionic bonds and hydrophobic interactions. This may result in destabilization of protein and increased formation of aggregations [54]. However, the aggregation may be dominated by the non-protein constituents as well, such as cellulose and hemicellulose, because 160–600 µm is too large in size and cellular matrixes have been observed from the SEM images (Figure 2). The hydrophobicity of these matrixes may be increased when washing with a higher water/ethanol ratio, and therefore they formed larger aggregates. In addition, the milling process after freeze drying may also have an effect. Washing with a high water/ethanol ratio resulted in the inclusion of more sugars in the SPCs (Figure 5), the higher sugar content may have led to a more ductile powder that was more difficult to mill, and therefore the size of particle remained large after dispersing in the water.

### 3.6. Nitrogen Solubility Index (NSI)

The protein solubility was determined as the nitrogen solubility index (NSI) and is presented in Figure 7. When the water/ethanol ratios were 60% or lower, the variations in NSI were small between the SPCs. Once the water/ethanol ratio increased from 60 to 80%, the solubility increased significantly. This outcome was consistent with other studies that have found that the protein solubility of ethanol-treated lentil protein isolate (LPI) was below 10% when 35–55% ethanol was applied, and increased to approximately 30% when the water/ethanol ratio increased from 55 to 75% [55]. Similarly, a higher NSI value for defatted soy meal was found when 100% ethanol was used for oil extraction as compared with 88% ethanol [56].

There are several possible explanations for this result. We found that the SPCs washed with 40% and 60% water/ethanol ratios presented the highest degree of protein denaturation among all the SPCs (Table 2), which might have resulted in more exposure of hydrophobic groups, and therefore relatively lower NSI values. The NSI of commercial SPC was approximately 20.41%, which was lower than the values of all the SPCs obtained in this study. This also correlates well with the observed complete denaturation of the commercial SPC. Another possible explanation for this is that at high water/ethanol ratios (80% and 100%), a higher proportion of water-soluble components (including proteins) remained in the solid phase, leading to higher NSI values. The result from the SDS-PAGE (Figure 3B) also showed the bands of protein with low molecular weight. These proteins are small and may contribute to the increase of overall protein solubility. Although the size of aggregates was more pronounced when the water/ethanol ratios were 60% and above (Figure 6), these aggregates were likely caused by the other constituents in the SPCs rather than soy protein.

### 3.7. Water Holding and Oil Absorption Capacity

The water holding capacity (WHC) and oil absorption capacity (OAC) are important functional properties for developing meat-related and meat analogue products [57,58]. Here, the WHC is defined as the amount of water bound per gram of SPC on a dry basis, while the WHC_P_ represents the amount of water bound per gram of insoluble pellet [20]. It was found that the WHC_P_ increased significantly when the water/ethanol ratio increased from 0 to 20%, and from 40 to 60% (Figure 8), which was consistent with the change of PSD (Figure 6). These insoluble aggregates can hold a certain amount of water as well. Furthermore, when the water/ethanol ratios were 60% and above, the WHC of the SPC was negatively correlated to the water/ethanol ratio. It seems that a decrease in WHC may have resulted from an increase in NSI (Figure 7), but the WHC_P_ and PSD were not influenced. This might confirm that the insoluble aggregates are dominated by the non-protein constituents such as cellulose and semi-cellulose, and therefore did not depend on a change in protein solubility. The WHC and WHC_P_ value of commercial SPC was 7.2 g water/g and 9.8 g water/g, respectively. These values were higher than the values of all the SPCs in this study. It has been reported that an additional heating treatment during the fractionation process can increase the WHC and WHC_P_ of soy protein, suggesting that the WHC of SPC can be further increased by incorporating a heating step in the washing process [59].

The OAC value of SPC was negatively affected by an increase in the water/ethanol ratio during the washing process, with SPCs washed with water presented the highest value of 1.3 g oil/g (Figure 8). A similar finding was also previously reported for the OAC of pea flour, which gradually deceased from 0.96 to 0.76 g oil/g when the alcohol concentration increased from 0 to 80% during an alcohol washing process [60]. In addition, this trend was consistent with the trend of oil left in the SPCs, that is, the lower the amount of oil left in the SPC, the lower the OAC value observed. In the case that high oil absorption is desired, water-only washing seems to be a good option.

### 3.8. Viscosity Analysis

For the viscosity analysis, the protein concentrations for all the SPC dispersions were standardized at 6 wt%, according to different protein contents in the SPCs (Table 1). With 6 wt% protein concentration, all the SPC samples were still fluid, based on a visual observation. When measured with a rheometer, a decrease in viscosity was observed with an increased shear rate, indicating that all the SPC dispersions exhibited similar shear-thinning behaviors under room temperature (Figure 9). The observed shear-thinning behaviors of all the SPCs were not influenced by the variations of particle size (Figure 6), which may imply that the shear thinning was caused by flow alignment more than shear-induced breakage of the smaller aggregates. Except for the SPC processed with water, the dispersions of all the other SPCs and commercial SPC exhibited similar viscosity values, despite the variations in the non-protein constituents in the fraction. This result suggests that the oil and carbohydrate contents have minor impacts on the viscosity of these mixtures. A previous study compared the viscosity of pea protein fractions with different protein purities, and reported that the viscosity was closely related to the protein of the fractions, while, indeed, the soluble and insoluble carbohydrates had limited impact [61]. Although a higher soy oil volume fraction, in the range of 5–35%, has been shown to lead to a higher viscosity of heated SPI emulsions [62], it is not expected that the oil content has a significant influence on the viscosity of non-heated mixtures. Therefore, the highest viscosity value of the water-only washed SPC may be mainly attributed to a change in soy protein. Most likely, this is a result of the fact that the average molecular weight of the proteins in the water-washed SPC is higher, as revealed by SDS-PAGE (Figure 3B).

### 3.9. Gelation Behaviors

The gelation behaviors of the SPCs upon and after thermal treatment were studied with temperature, frequency, and strain sweep applications, and are shown in Figure 10. The protein concentrations for all the samples were standardized at 9 wt%, according to different protein contents in the SPCs (Table 1). For all the SPC samples, the storage module (G′) values were constantly higher than loss module (G″) values, even without heating. This indicated that a dispersion with 9 wt% protein concentration was able to form solid-like structures at room temperature. During the temperature sweep, similar G′ and G″ value changes were observed for all the SPC samples. Abrupt increases in the G′ and G″ values were found by heating the sample at a temperature range from 70 to 95 °C, which may have been caused by gel formation. During the constant temperature regime at 95 °C, the value of the G′ increased somewhat, while the G″ value was almost stable. After that, a slight decrease followed for both G′ and G″. During cooling, the G′ and G″ values both increased significantly for all the SPCs, indicating that the gel structures were further improved by the formation of additional bonds [63]. The commercial SPC showed similar gelation behaviors under the same measurements. When considering the values of G′ and G″ obtained from different SPCs, the SPC processed with 0% ethanol had the highest values before heating, which was similar to the finding of viscosity. However, for the SPCs processed with 80% and 100% water/ethanol ratios, the G′ and G″ values increased more pronounced than other SPCs upon heating and cooling. The final G′ and G″ values of these two SPC samples were comparable to the SPC processed with 0% ethanol after the temperature sweep.

During the frequency sweep, the G′ and G″ values of all the SPCs were relatively constant over the frequency range of 0.1–10 Hz. The highest values were found for SPCs processed with 0%, 80%, and 100% water/ethanol ratios, which were in the range of 1000–10,000 Pa, approximately 10 times higher as compared with SPCs processed with 40% and 60% ethanol. The high value of water-washed SPC may have been caused by the formation of lipid-protein complex after heating, since washing with water hardly remove any oil from the DFSM. For the high ethanol-washed (80% and 100%) SPCs, the presence of low-molecular-weight proteins may work actively and contribute to the high G′ and G″ values. The 2S albumins are a group of storage proteins in soybean with molecular weights of approximately 18 kDa. They contain thiol groups that could lead to disulfide bridge formation at heating stage [64].

The length of the linear viscoelastic (LVE) regime of all the SPCs was studied using a strain sweep at constant frequency afterward. For all SPC gels, a linear response was observed, wherein the G′ and G″ values were independent of strain. When the strain was further increased, the G′ and G″ values decreased sharply, suggesting breakage of bonds in the gel network and transition from linear to non-strain behavior [65]. The lengths of the LVE regimes of SPC gels (80% and 100% ethanol washed) were rather similar, and longer than other SPCs, which could be related to higher gel strength [66].

### 3.10. Textural Analysis of Soy Gels

The water/ethanol ratio during the washing process affected the soy gel textures, as shown in Figure 11. Firm gels can be formed by all the SPCs after heating with a standardized 15 wt% protein concentration. The textural properties of the gels formed by all the SPCs were comparable to the gel formed when using commercial SPC, with the exception of the water-only washed SPC, which showed the highest hardness, chewiness, and cohesiveness among all the SPCs. The remarkable high values of hardness, chewiness, and cohesiveness are probably related to the high molecular weight of the protein present. In addition, the high residual oil in the SPCs might play a role. It is known that various lipid substances have impacts on the gelation of protein. Previous studies have reported that soybean oil addition (1–5%) enhanced the hardness and cohesiveness of heat-induced SPI gels under both 20% and 60% deformation (Miura and Yamauchi, 1984). The increasing hardness with increasing oil content in the whole soybean protein gel was also described by Yamamo et al. [67].

In the previous section, the SPC gels (80% and 100% ethanol washed) presented the highest G′ values as well as the longest length of the LVE regimes among all the gels after heating, which could be related to a higher gel strength. In contrast to earlier findings, however, in this study, no evidence was found that the gel characteristics of 80% and 100% washed SPC gels were more prominent than others. A possible explanation might be that the higher gel strength was within the small deformation range, and correlated to the displacement of protein aggregates or other components, while the gel characteristics were determined within the large deformation range, referring to the disruption of a gel network [68].

## 4. Conclusions

This research examined the effect of aqueous ethanol washing on cold-pressed DFSM. Different water/ethanol ratios were used to investigate the effect of ethanol on the overall composition, yield, and functionality of SPCs. According to the results, the aqueous ethanol washing process enhanced the protein content in SPC, while all the obtained SPCs after the washing process still contained oil. Water-washed (0% ethanol) SPC, however, presented increased oil content with low oil oxidation, and the best OAC, while the water washing treatment seemed to obtain a relatively native ingredient. With regards to the rheological properties, the water-only washed SPC dispersion had the highest viscosity and formed the gel with the highest gel strength and hardness. An additional advantage of using water-only washing after cold pressing is that it is a completely solvent-free process. Among all the water/ethanol ratios, washing with high ethanol level (60% and above) led to SPCs with low oil content and high yield. These SPCs also presented a low degree of denaturation, better protein solubility, and higher water holding capacity.

Overall, it can be concluded that the water/ethanol ratio is an interesting parameter for controlling the protein composition, yield, and functional properties of the SPC. The most interesting functional properties were obtained when using either water-only solvent or solvent with high water/ethanol ratios. When using moderate ethanol concentrations, less favorable properties were found, including high oil oxidation levels. Nevertheless, soy protein ingredients can be produced with a comparable composition to commercial SPC, but with a range of functionaries that are relevant for the development of novel soy-food applications.

## Figures and Tables

**Figure 1 foods-10-02222-f001:**
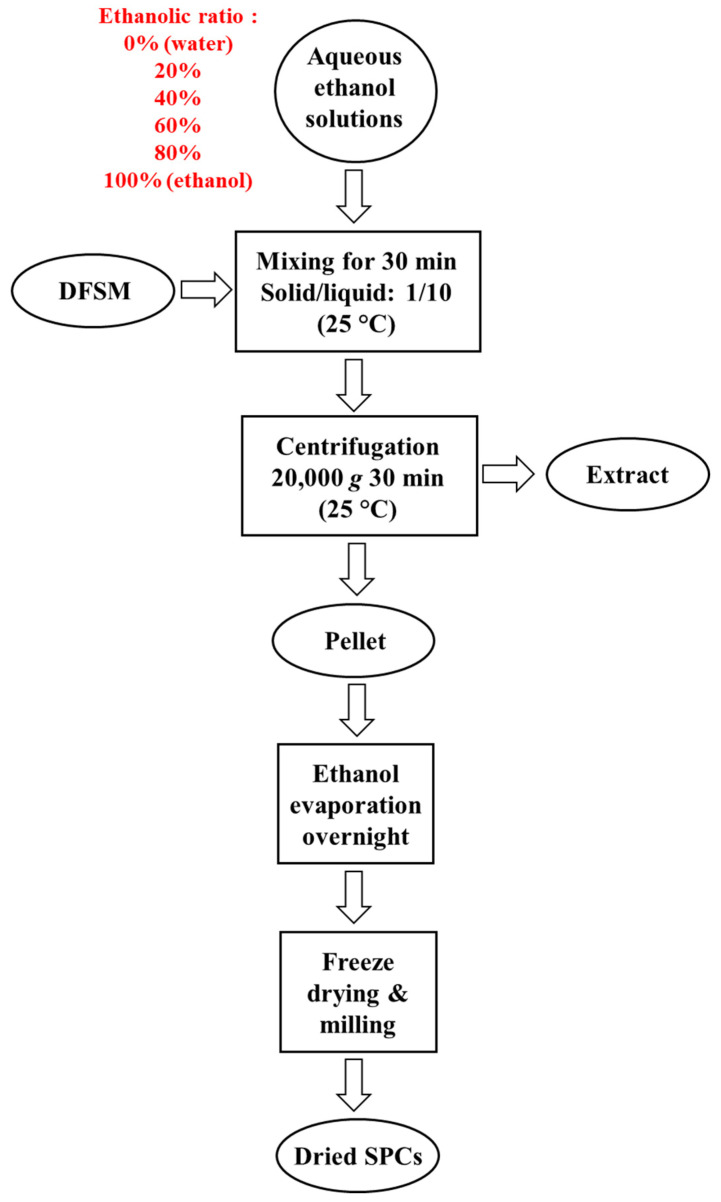
Scheme for the aqueous ethanol washing process.

**Figure 2 foods-10-02222-f002:**
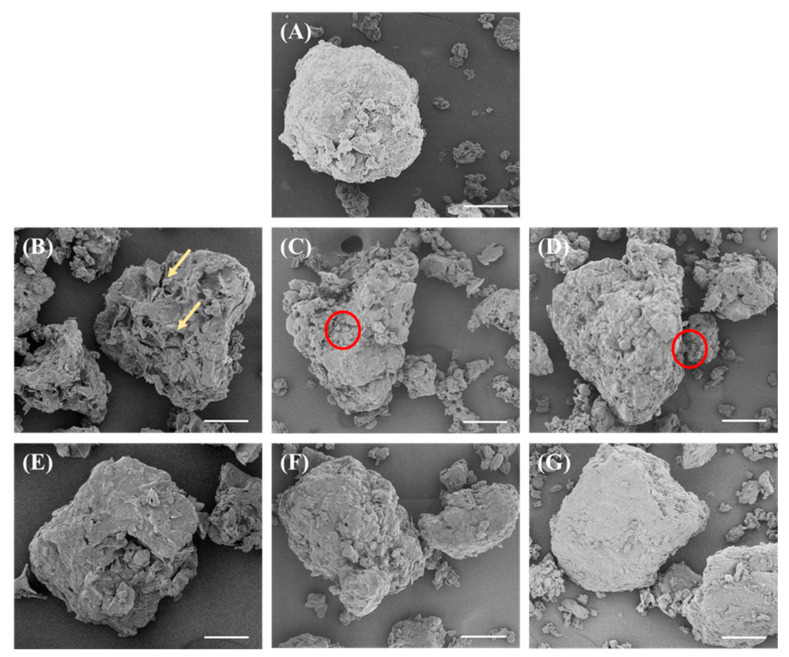
SEM images for defatted soy meal (DFSM) (**A**) and aqueous ethanol-washed soy protein concentrates (SPCs) with water/ethanol ratios of 0% (**B**); 20% (**C**); 40% (**D**); 60% (**E**); 80% (**F**); and 100% (**G**). The scale bar is 50 µm. The yellow arrow and red circle represent cellular matrix and protein body, respectively.

**Figure 3 foods-10-02222-f003:**
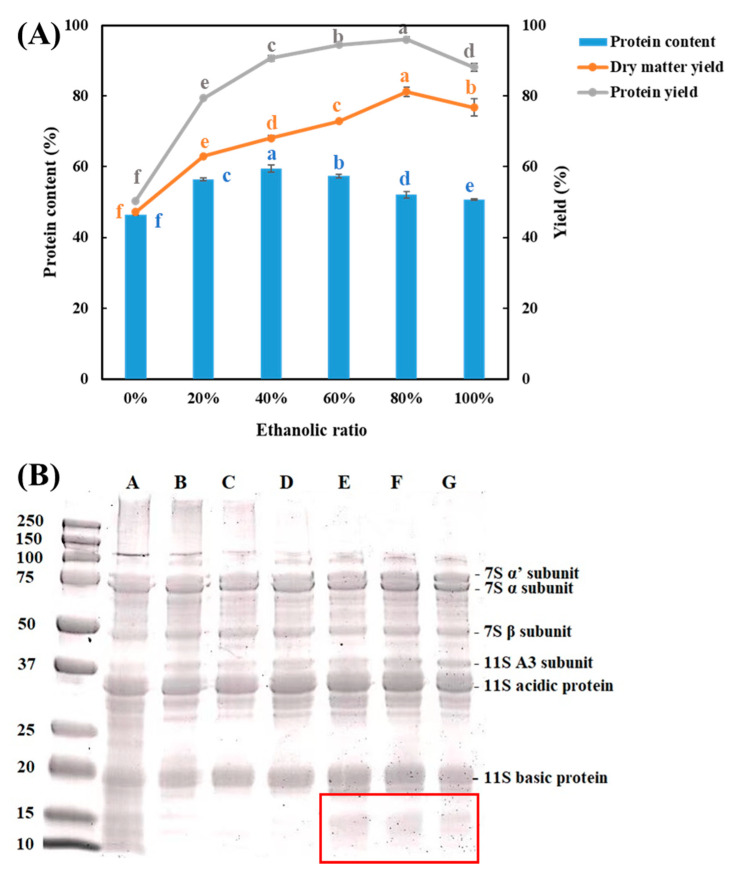
(**A**) Protein content, protein yield, and dry matter yield of all the soy protein concentrates (SPCs). The values in the figure are compared in series (same color) and different top letters (a,b,c,d,e,f) indicate a significant difference (*p* < 0.05); (**B**) SDS-PAGE electrophoresis of soluble proteins in soy samples. Lane A, commercial SPC; Lane B–G, SPCs obtained by washing process with the water/ethanol ratios of 0%, 20%, 40%, 60%, 80%, and 100%.

**Figure 4 foods-10-02222-f004:**
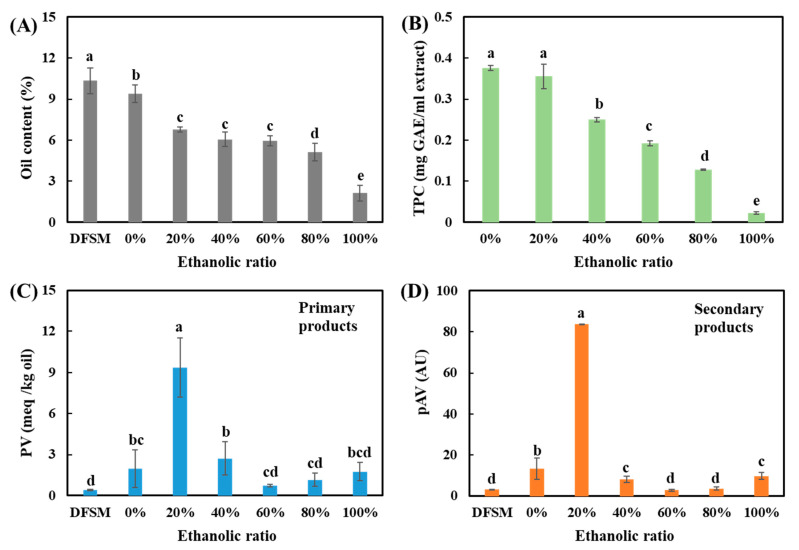
(**A**) Oil content of defatted soy meal (DFSM) and all the soy protein concentrates (SPCs); (**B**) total phenolic content (TPC) of all the extracts; (**C**) hydroperoxide concentration (PV); (**D**) *p*-anisidine value (pAV) of soybean oil contained in DFSM and SPCs. The values in the figure are compared and different top letters (a,b,c,d,e) indicate a significant difference (*p* < 0.05).

**Figure 5 foods-10-02222-f005:**
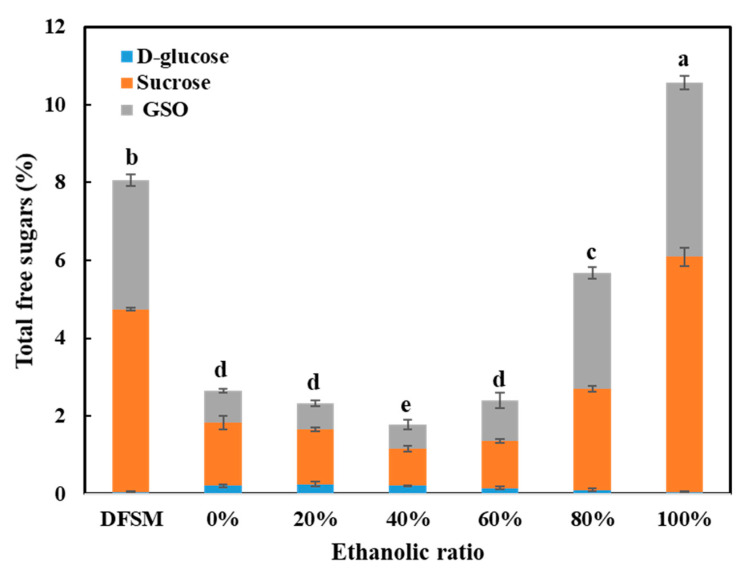
Total free sugars (D-glucose, sucrose, and GSO) in the defatted soy meal (DFSM) and all the soy protein concentrates (SPCs). The values in the figure are compared and different top letters (a,b,c,d,e) indicate a significant difference (*p* < 0.05).

**Figure 6 foods-10-02222-f006:**
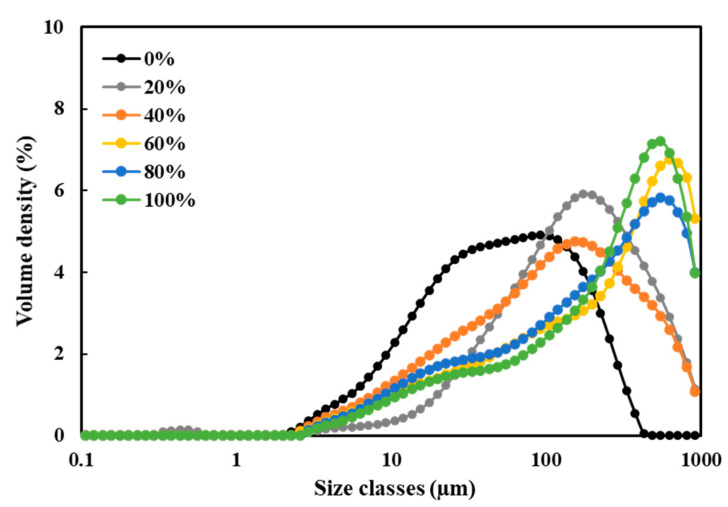
The particle size distribution of 1% (*w*/*v*) soy protein concentrate (SPC) dispersions determined by a Mastersizer.

**Figure 7 foods-10-02222-f007:**
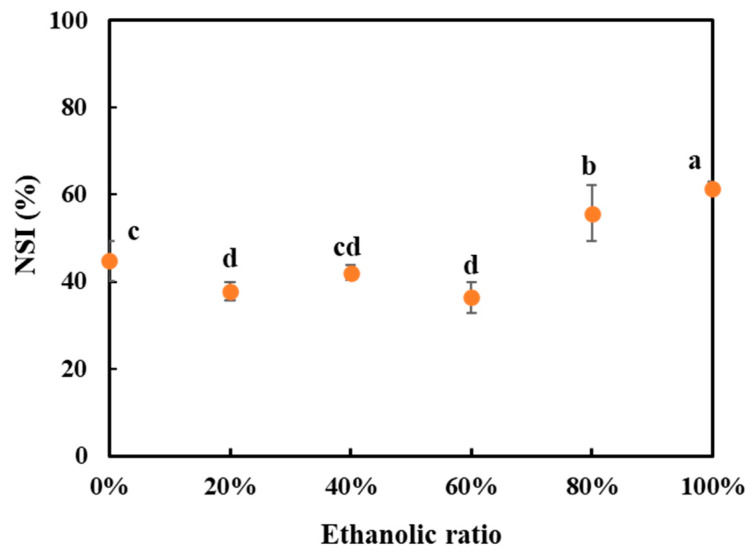
The nitrogen solubility index (NSI) of all the soy protein concentrates (SPCs). The values in the figure are compared and different top letters (a,b,c,d,e) indicate a significant difference (*p* < 0.05).

**Figure 8 foods-10-02222-f008:**
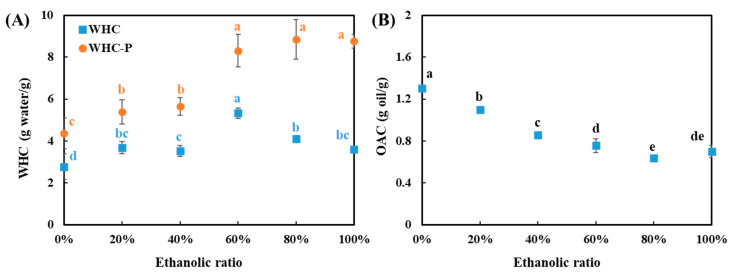
(**A**) Water holding capacities (WHC and WHC_P_), and (**B**) oil absorption capacity (OAC) of all soy protein concentrates (SPCs). The values in the figure are compared in series (same color) and different top letters (a,b,c,d,e) indicate a significant difference (*p* < 0.05).

**Figure 9 foods-10-02222-f009:**
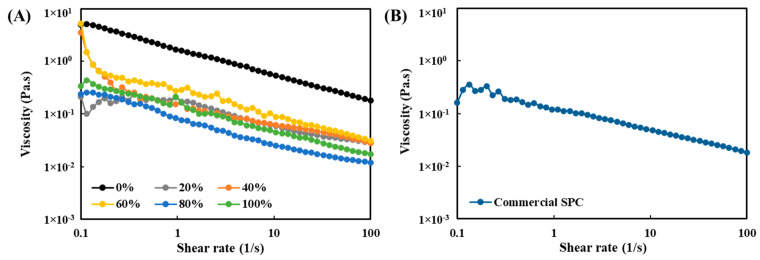
Viscosity as a function of shear rate at 25 °C of (**A**) aqueous ethanol-washed soy protein concentrates (SPCs); and (**B**) commercial SPC dispersions (standardized 6 wt% protein content).

**Figure 10 foods-10-02222-f010:**
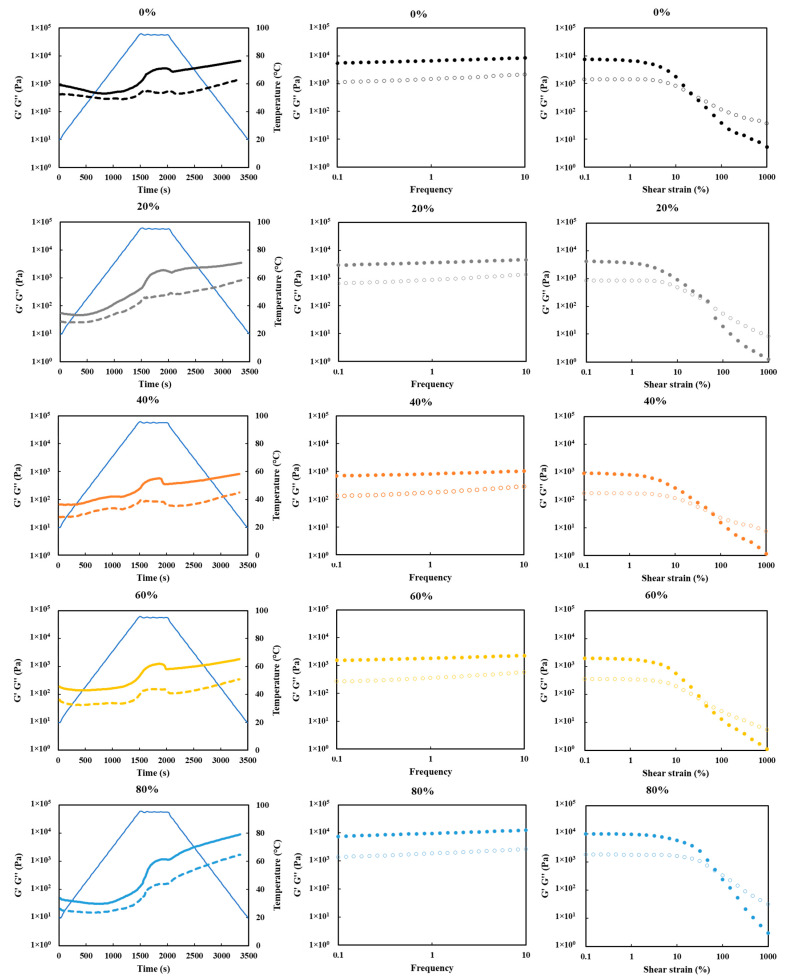
Temperature (**left**), frequency (**middle**), and strain (**right**) sweeps sequentially applied on all the soy protein concentrate (SPC) samples (standardized 9 wt% protein content). G′, closed symbols; G″, open symbols; temperature, solid blue line.

**Figure 11 foods-10-02222-f011:**
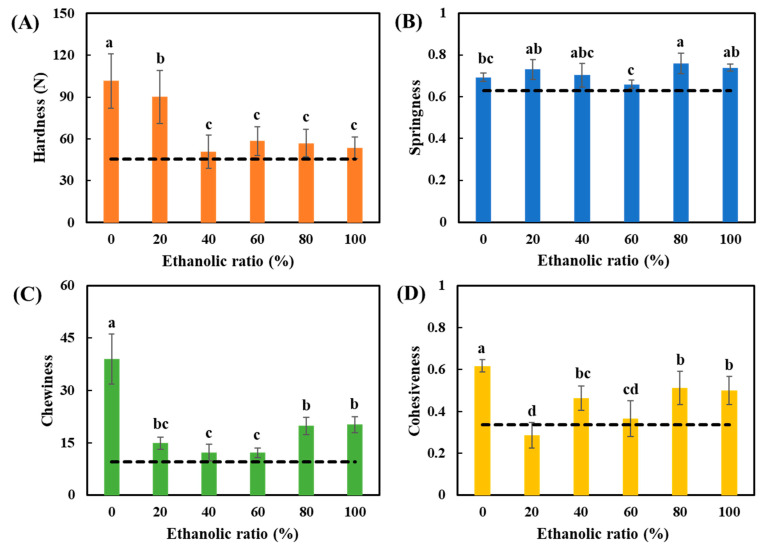
Textural analysis on (**A**) Hardness, (**B**) Springiness, (**C**) Chewiness and (**D**) Cohesiveness of all the soy gels (standardized 15 wt% protein content). The black square dotted lines represent the reference value measured from commercial SPC gels. The values in the figure are compared and different top letters (a,b,c,d) indicate a significant difference (*p* < 0.05).

**Table 1 foods-10-02222-t001:** Sample preparation to obtain the soy protein concentrate (SPC) samples with 6, 9, and 15 wt% protein concentrations.

Water/Ethanol Ratios	Protein Content of SPC, db.%	Solid Content of the SPC Samples, %		
		6 wt% Protein Concentration	9 wt% Protein Concentration	15 wt% Protein Concentration
0%	46.48 ± 0.29	12.91 ± 0.08	19.36 ± 0.12	32.27 ± 0.20
20%	56.43 ± 0.47	10.63 ± 0.09	15.95 ± 0.13	26.58 ± 0.22
40%	59.49 ± 1.11	10.09 ± 0.19	15.13 ± 0.28	25.22 ± 0.47
60%	57.36 ± 0.60	10.46 ± 0.11	15.69 ± 0.16	26.15 ± 0.27
80%	52.10 ± 0.92	11.52 ± 0.20	17.27 ± 0.31	28.79 ± 0.51
100%	50.70 ± 0.21	11.83 ± 0.05	17.75 ± 0.07	29.58 ± 0.12
Commercial SPC	61.68 ± 0.00	9.73 ± 0.00	14.59 ± 0.00	24.32 ± 0.00

**Table 2 foods-10-02222-t002:** The denaturation temperature and enthalpy of the transition of the soy protein concentrates (SPCs).

Water/Ethanol Ratios	7S T_d_ (°C)	Enthalpy (J/g)	11S T_d_ (°C)	Enthalpy (J/g)
0%	75.72 ± 0.44 a	1.73 ± 0.36 a	94.30 ± 0.21 d	3.31 ± 0.31 ab
20%	75.20 ± 0.19 a	1.85 ± 0.12 a	95.05 ± 0.04 c	3.59 ± 0.64 ab
40%	75.94 ± 1.45 a	0.70 ± 0.04 c	92.44 ± 0.19 e	3.98 ± 0.74 a
60%	74.64 ± 1.56 a	0.75 ± 0.13 c	92.74 ± 0.49 e	3.38 ± 0.27 ab
80%	76.11 ± 0.33 a	1.33 ± 0.07 b	95.76 ± 0.09 b	3.48 ± 0.14 ab
100%	76.08 ± 0.14 a	1.19 ± 0.14 b	96.73 ± 0.27 a	3.14 ± 0.31 b

The values in the table are compared per each column and different top letters (a,b,c,d,e) indicate a significant difference (*p* < 0.05).

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
