# Peer review of "Characteristics of Soy Protein Prepared Using an Aqueous Ethanol Washing Process"

_foods, 2021, doi:10.3390/foods10092222_

Round 1
Reviewer 1 Report
Manuscript titled "Characteristics of soy protein prepared by aqueous ethanol washing process" by Peng et al decribed body of work which is currently very topical. It is generally accepted that major hindrance to wider application of plant protein is associated specific flavour &/or off-flavour, which is primarily caused by the oxidative deterioration of residual lipid. The authors in the above study present elegantly designed aqueous ethanol washing procedure to reduced residual fat in plant protein fraction. The plant protein material thus prepared was well characterized for chemical and physical properties. The manuscript is very well written and laid out. I would have minor modifications to figures (Figures 1, 3a, 4 and 5- writing appear to be quite fuzzy and Figures 7 & 10 - black background is not helpful, recommend clear background).
Reviewer 2 Report
This manuscript describes the effect of aqueous ethanol washing on cold-pressed soybean meals. This paper is interesting and could extend the knowledge in our field. Detailed comments are shown below.
1) The authors mentioned that de-fat soy meal (DFSM) is consist of 43% protein and 10% oil. The other main component should be non-starch polysaccharides such as cellulose, and their amount could be equal to or greater than the amount of protein. The authors added an aqueous ethanol washing process at different concentrations and investigated. Actually, we call soy protein concentrate (SPC) when soy flour is washed with ethanol. This is not the case. Thus, I suggest that the authors should take this into account and revise properly, such as SPFs (L16, L68), soy protein ingredients (L56), etc.
2) Figures are necessary to revise properly. The resolution of Fig. 1 is too low. The background in Fig. 7 and 10 is black. Is not preferable. Most of titles and scales in axis are difficult to see, Figs. 3A, 4, 6 etc. Fig. 3B seems to be a copy, they need to change to a proper image. Only abbreviations in figure legends are not acceptable, DFMS, SPFs in Fig. 2, etc.
L123 RAFGL assay: RAFGL is an abbreviation for Raffinose Sucrose D-Glucose. Please, revise.
L131-135: Why did the authors examined soluble protein but not all protein? They need to explain in the text.
L139 Mini-protein gel: Is this the product name? We use polyacrylamide, but not protein, gels to separate proteins. Please, revise.
L332: Why can the authors mention ‘acceptable’? Please explain in the text.
L381-405 7S is denatured but 11S is not. I think their explanation is not make sense and need to discuss properly.
L407-433: I think that PSD did not measure protein particles or aggregates because 160-600 micron meter is too large in size. These could be mainly due to cell walls.
L562: Fig. 11 is missing.
Round 2
Reviewer 2 Report
The manuscript was revised properly according to the reviewer's comment.